# Yttrium Doping Effects on Ferroelectricity and Electric Properties of As-Deposited Hf_1−x_Zr_x_O_2_ Thin Films via Atomic Layer Deposition

**DOI:** 10.3390/nano13152187

**Published:** 2023-07-27

**Authors:** Youkyoung Oh, Seung Won Lee, Jeong-Hun Choi, Seung-Eon Ahn, Hyo-Bae Kim, Ji-Hoon Ahn

**Affiliations:** 1Department of Materials Science and Chemical Engineering, Hanyang University, Ansan 15588, Republic of Korea; sponge0520@naver.com (Y.O.); wehou1103@naver.com (S.W.L.); wkdqhrh77@gmail.com (J.-H.C.); 2Department of Nano & Semiconductor Engineering, Tech University of Korea, Siheung 15073, Republic of Korea; seahn@tukorea.ac.ar

**Keywords:** atomic layer deposition, Hf_1−x_Zr_x_O_2_, yttrium, crystalline phase, ferroelectric, dielectric

## Abstract

Hf_1−x_Zr_x_O_2_ (HZO) thin films are versatile materials suitable for advanced ferroelectric semiconductor devices. Previous studies have shown that the ferroelectricity of HZO thin films can be stabilized by doping them with group III elements at low concentrations. While doping with Y improves the ferroelectric properties, there has been limited research on Y-HZO thin films fabricated using atomic layer deposition (ALD). In this study, we investigated the effects of Y-doping cycles on the ferroelectric and electrical properties of as-deposited Y-HZO thin films with varying compositions fabricated through ALD. The Y-HZO thin films were stably crystallized without the need for post-thermal treatment and exhibited transition behavior depending on the Y-doping cycle and initial composition ratio of the HZO thin films. These Y-HZO thin films offer several advantages, including enhanced dielectric constant, leakage current density, and improved endurance. Moreover, the optimized Y-doping cycle induced a phase transformation that resulted in Y-HZO thin films with improved ferroelectric properties, exhibiting stable behavior without fatigue for up to 10^10^ cycles. These as-deposited Y-HZO thin films show promise for applications in semiconductor devices that require high ferroelectric properties, excellent electrical properties, and reliable performance with a low thermal budget.

## 1. Introduction

HfO_2_-based ferroelectric thin films have attracted significant attention due to their exceptional ferroelectric properties even at sub-10 nm thickness. These films also have advantages such as wide bandgap, CMOS-compatible storage technologies, and their lead-free nature, compared to conventional ferroelectrics such as Pb(Ti,Zr)O_3_ [1,2,3]. Among the various HfO_2_-based ferroelectric thin films, Hf_1−x_Zr_x_O_2_ (HZO) thin films are particularly promising due to their property to exhibit ferroelectricity at lower crystallization temperature (400–600 °C), and the robust remnant polarization, making them a favorable choice for back-end-of-line (BEOL) integration [1]. As reported, a non-centrosymmetric polar orthorhombic phase (o-phase, Pca2_1_) is responsible for ferroelectricity in HfO_2_-based thin films [4]. The o-phase in HfO_2_-based thin films can be stabilized by doping with group III dopants, such as Al, La, and Y, which form oxygen vacancies, thereby reducing the energy barrier of the ferroelectric phase [5]. Among them, ferroelectric properties for Y are greatly improved by doping with HfO_2_-based thin films [6]. Various studies have been conducted on the effect of group III dopants on HfO_2_ thin films using various deposition methods [7,8,9]. More recently, studies focused on the effects of group-III dopants with low doping concentrations on the ferroelectric properties of HZO thin films. Group III dopants contribute not only to the stabilization of the o-phase but also to the stabilization of the tetragonal (t-phase) and cubic (c-phase) phases depending on their doping concentration, demanding precise control of the doping concentration at fine scales [5,7,10]. Thus, the necessity for atomic layer deposition (ALD) processes with excellent controllability for low-level doping, thickness, and conformal integration has become more pronounced [11]. To date, there have been some reports on ferroelectric Al-HZO and La-HZO ferroelectric thin films deposited by ALD; however, there have been only a few reports on Y-HZO ferroelectric films prepared by ALD [10,12,13,14]. Most Y-HZO films have been deposited using the chemical solution deposition (CSD) method, which is limited to controlling fine doping concentrations at small thicknesses and requires very high annealing temperatures [11,15,16]. Meanwhile, as mentioned, a low thermal budget is essential for integrating ferroelectric devices into the BEOL process, which has a limited temperature below 400 °C [17,18]. From this perspective, we reported the first superior and stable ferroelectricity of HZO thin films deposited by ALD using a thermally stable cyclopentadienyl (Cp)-based cocktail precursor without annealing [19]. In addition, we systematically investigated the ferroelectric behavior of the as-deposited HZO thin films with various Zr compositions. The transition behaviors of paraelectric, ferroelectric, and antiferroelectric characteristics were observed for the first time in the as-deposited state [20].

Building on our previous research, this study focused on investigating the effects of Y-doping cycles on the ferroelectric and additional electrical properties of crystalline 10-nm-thick HZO thin films fabricated via ALD without post-thermal process. At low Y-doping cycles, the ferroelectric properties of the as-deposited Y-HZO thin films were enhanced by the phase transition to the o-phase. As the number of Y-doping cycles increased, the o-phase, t-phase, and c-phase gradually stabilized, and the ferroelectric transition behavior (ferroelectric–antiferroelectric–paraelectric) of the as-deposited Y-HZO thin films was observed. Moreover, the boundary between the ferroelectric- and antiferroelectric-like properties of the Y-HZO thin films can be changed by changing the initial composition of the HZO thin films. It is important to note that, compared to the Y-doped films that required high-temperature annealing, the films deposited in this study have various phase transitions depending on the Y concentration without post-thermal process. In addition, the as-deposited Y-HZO thin films exhibited several advantages, including an improved dielectric constant, reduced leakage current density, enhanced endurance, and decreased coercive field value compared to pure HZO thin films. These desirable characteristics make the as-deposited Y-HZO thin films suitable for various semiconductor devices that require high ferroelectric properties, excellent electrical properties, and reliability within a low thermal budget. Examples of such applications include capacitor-based ferroelectric random-access memories (FRAM), ferroelectric field-effect transistors (FeFETs), and ferroelectric tunnel junctions (FTJs) [21,22].

## 2. Materials and Methods

### 2.1. Sample Fabrication Process

The Y-HZO (yttrium-doped Hf_1−x_Zr_X_O_2_) thin films were deposited using ALD in a shower-head type reactor (iOV d150, iSAC RESEARCH Co., Ltd., Republic of Korea) on TiN (200 nm)/SiO_2_/Si substrates. Hf[Cp(NMe_2_)]_3_ (HAC, iChems Co., Ltd., Republic of Korea), Zr[Cp(Nme_2_)]_3_ (ZAC, iChems Co., Ltd., Republic of Korea), and a newly synthesized precursor, (RCp)_2_Y(^i^Pr-amd) (IYA02, iChems Co., Ltd., Republic of Korea) were used as Hf, Zr, and Y precursors, respectively, along with ozone (O_3_, 220 g/m^3^) as an oxidant. The deposition process was carried out at a temperature of 320 °C to ensure ferroelectric phase (non-centrosymmetric orthorhombic phase, Pca2_1_)-dominated crystallinity without a subsequent thermal treatment process (as-deposited ferroelectric HZO thin films) [19]. To control the composition of the Y-HZO thin films, we adopted an ALD super-cycle, as shown in Figure 1a. The length and height of the schematic in Figure 1a were proportional to the pressure and time, respectively. To investigate the effects of the initial composition of the HZO thin films on the Y doping, the ‘m’/’n’ ratios for depositing HfO_2_ and ZrO_2_ consisting of one HZO super-cycle were adjusted to 3:2 (Hf_0.6_Zr_0.4_O_2_, H6Z4), 1:1 (Hf_0.5_Zr_0.5_O_2_, H5Z5), and 2:3 (Hf_0.4_Zr_0.6_O_2_, H4Z6). The Y-HZO super-cycle was repeated ‘p’ times to match the required Y-doping cycle. Also, deposition was performed by adjusting the number of repetitions of the HZO super-cycle to ensure that the Y_2_O_3_ doping cycles were equally distributed in the middle of the HZO thin films [23]. As mentioned above, because the ALD process induces crystallization during the deposition process, an additional post-thermal treatment process for crystallization was not performed on all thin films. To evaluate the electrical properties, metal-ferroelectric-metal (MFM) capacitors were fabricated by sputtering 100-nm-thick TiN top electrodes using a shadow mask with a radius of 100 μm.

### 2.2. Characterizations

The thicknesses of the as-deposited Y-HZO thin films were measured using spectroscopic ellipsometry (SE MG-1000, Nano View, Republic of Korea), and their chemical compositions were analyzed using X-ray photoelectron spectroscopy (XPS, K-alpha, Thermo Scientific). Grazing-angle incidence X-ray diffraction (GI-XRD, SmartLab, Rigaku, Japan) using Cu-Kα radiation (λ = 1.5405 Å) with an incidence angle of 1° and high-resolution transmission electron microscopy (HR-TEM, JEM-3011 HR, JEOL Co., Ltd., Japan) were performed for crystal structure analysis of the thin films. To estimate the electrical properties of the thin films, the polarization-voltage (*P-V*) curves, capacitance-voltage (*C-V*) curves, and current-voltage (*I-V*) curves were obtained using semiconductor parameter analysis equipment (4200-SCS, Keithley, USA) with a 4225 PMU module. Endurance tests were conducted to determine the reliability of the MFM capacitors.

## 3. Results

### 3.1. ALD Characteristics of Y_2_O_3_ Thin Films

Figure 1b–d display the ALD characteristics of Y_2_O_3_ thin films for Y-HZO thin film deposition. The ALD process temperature window with Y_2_O_3_ thin films using IYA02 precursor was characterized by a change in growth per cycle (GPC) as a function of deposition temperature from 260 to 370 °C, as shown in Figure 1b. The Y_2_O_3_ thin films were deposited stably with GPC of about 0.2 nm/cycle at temperature range of 320 to 360 °C, and these results indicate that the IYA02 precursor can be used as a dopant for the HZO thin film at a deposition temperature of 320 °C. Figure 1c,d elucidate GPC of the Y_2_O_3_ thin films at a deposition temperature of 320 °C as a function of IYA02 precursor dosing time and purge time, respectively. In Figure 1c, it can be observed that the GPC of Y_2_O_3_ thin films gradually increased and saturated at approximately 0.195 nm/cycle after 18 s of precursor dosing time, showing ALD self-saturation growth behavior. Also, as shown in Figure 1d, the self-saturation behavior according to the precursor purge time was also confirmed. The stable purge time of 30 s was set to ensure that any residual products were sufficiently removed. Therefore, the IYA02 precursor dosing time and purge time were set at 18 s and 30 s, respectively, for the Y-doping cycle in HZO thin films [20].

### 3.2. Chemical Composition of As-Deposited Y-HZO Thin Films

XPS analysis was performed to investigate the correlation between the deposition cycle ratio and the composition of metal atoms in the HZO and Y-HZO thin films [24]. Figure 2a shows the XPS atomic ratios of Hf and Zr as functions of the Zr/(Hf + Zr) cycle ratio for the undoped HZO thin films. The atomic ratio of Hf and Zr was obtained by analyzing the peak spectra of Hf4f and Zr3d of the Y-HZO thin film, and each ratio was calculated as atomic percent of Hf/(atomic percent of Hf + atomic percent of Zr), atomic percent of Zr/(atomic percent of Hf + atomic percent of Zr), as shown in Appendix A. When the Zr/(Hf + Zr) cycle ratios were 0.4, 0.5, and 0.6, the atomic ratios of Zr/(Hf + Zr) were approximately 0.4, 0.5, and 0.6, respectively, similar to the cycle ratios. These results confirmed that the Hf and Zr compositions of the HZO thin films were successfully controlled by the ALD super-cycle. Based on these results, the Y atomic ratios measured by XPS as a function of the number of Y-doping cycles in the Y-H6Z4 and Y-H4Z6 thin films are shown in Figure 2b. Regardless of the composition of the HZO matrix thin films, a gradual increase in the Y atomic ratio was observed with increasing Y-doping cycles, and the Y-doping cycle was 1, 2, 4, 7, and 10, the Y content in the Y-doped HZO thin film was observed to be 0.004, 0.009, 0.019, 0.064, and 0.078, respectively. This implies that not only the Hf and Zr concentrations but also the fine Y-doping concentrations in the Y-HZO films can be precisely controlled through the ALD cycle design (shown in Appendix A).

### 3.3. Crystallinity of As-Deposited Y-HZO Thin Films

GI-XRD analysis was performed to assess the crystallinity of the as-deposited Y-HZO thin films. The GI-XRD patterns of the Y-H6Z4 thin films after 0, 1, 2, 4, 7, and 10 Y-doping cycles are shown in Figure 3a, while those of the Y-H4Z6 thin films are shown in Figure 3b (GI-XRD spectra of as-deposited Y-H6Z4, Y-H5Z5, Y-H4Z6, and Y-H2Z8 thin films were shown in Appendix A). The observation of peaks in the GI-XRD patterns for both compositions (H6Z4 and H4Z6) indicates that Y-HZO thin films were crystallized during the deposition process at 320 °C. In the 2θ range from 26.5° to 32.5°, peaks were near 28.5° and 30.5°, which indicate monoclinic (m-phase) (−111 m) and o/t/c-phase (111 o/011 t/111 c), respectively. As the number of Y-doping cycles increased in both compositions of the Y-HZO thin films, the peak intensity in the m-phase decreased, whereas the peak intensity in the o/t/c-phase significantly increased [25]. These results are consistent with those of previous research, indicating that Y doping suppresses the m-phase and stabilizes the o/t/c-phase [26]. This suggests that the ferroelectric properties of Y-HZO thin films induce a reduction in the m-phase and stabilization of the o/t/c phase in a specific Y-doping range. Meanwhile, the polarization characteristics of the Y-HZO thin films could be transformed because of the phase transition within the o/t/c-phase; however, it was difficult to distinguish the exact phase between the o/t/c-phases located at approximately 30.5° in the GI-XRD pattern. Therefore, *P-V* curve and *C-V* curve measurements were subsequently conducted to determine the change in the polarization properties of the Y-HZO thin films due to the phase transition, as shown in Figure 4 and Figure 5 [16,27]. Figure 3c shows the variation in the intensity ratio of the m-phase peak and o/t/c-phase peak for the Y-HZO thin films with respect to the Y-doping cycle (details for peak deconvolution were shown in Appendix A). In the Y-H6Z4 thin films, the peak intensity ratio of the o/t/c phase exhibited a relatively high rate of increase about 1.7 times, ranging from 59% to 100%, whereas the Y-H4Z6 thin films showed an increase from 69% to 100%, with a relatively low rate of approximately 1.4 times. This result was attributed to the initial phase difference according to the Hf-Zr content in the undoped HZO thin films, and is expected to affect the polarization behavior of the Y-HZO thin films according to the Y-doping cycle (shown in Appendix A) [26]. To further investigate the change in crystallinity induced by Y doping in the HZO thin films, the crystallinity of the Y-H6Z4 thin films with a relatively high o/t/c-phase peak intensity ratio was analyzed using cross-sectional TEM. The analysis focused on two cycles of Y doping, where the o/t/c-phase peak intensity ratio increased to more than 90%, to evaluate the crystallinity of the films before and after Y doping. In the pure H6Z4 thin film shown in Figure 3d, the ferroelectric o-phase existed, but an m-phase that interfered with the ferroelectric properties was also present [28]. In contrast, for the Y-H6Z4 thin film with two cycles of Y doping, as shown in Figure 3e, the m-phase almost disappeared, and other phases, including the o-phase, significantly increased and were evenly distributed throughout the film. These results are in line with the results of the XRD analysis and successfully confirm the potential of improving the ferroelectric properties of HZO thin films through Y doping.

### 3.4. Electrical Properties of As-Deposited Y-HZO Thin Films

*P-V* measurements were performed to observe the polarization behavior according to the phase transition affected by the Y-doping cycle and the initial composition of the HZO thin films. Figure 4a shows the *P-V* curves as a function of the Y-doping cycle for the as-deposited Y-HZO thin films of various compositions. The as-deposited HZO thin films exhibited ferroelectric properties, even though Y was not doped at any composition, which increased as the Zr content increased from H6Z4 to H4Z6, as reported in a previous study [20]. The properties of these thin films gradually changed as the Y-doping amount increased. Until a certain level of Y doping, the ferroelectricity improved owing to the stabilization of the o-phase, which was expected to be due to the formation of oxygen vacancies. This can be easily understood from other study that the o-phase becomes more favorable in terms of total energy compared to the m-phase as the oxygen vacancy concentration increases [17]. After a certain level, the Y-HZO thin films exhibited antiferroelectric-like properties owing to the stabilization of the t-phase. After 10 cycles of Y doping, all Y-HZO thin films showed paraelectric properties induced by the c-phase, which stabilized after the o/t phase was sufficiently stabilized [7]. Thus, the ferroelectric, antiferroelectric-like, and paraelectric behaviors of the as-deposited Y-HZO thin films were established with the stabilization of the o-, t-, and c-phases depending on the Y-doping cycle. Figure 4b shows the variation in 2P_r_ (remanent polarization) values with the number of Y-doping cycles for HZO thin films of different compositions. The maximum increase in the 2P_r_ value was achieved at 4 Y-doping cycles for Y-H6Z4 thin films, 2 for Y-H5Z5 thin films and Y-H4Z6 thin films, and 3.3, 2.17, and 1.54 times, respectively. In addition, the higher the Zr content in the initial HZO thin films, the more changes in *P-V* curves from ferroelectric to antiferroelectric-like properties, manifested as a decrease in the 2P_r_ value, were observed at a relatively low Y-doping cycle. This can be interpreted as earlier stabilization of the antiferroelectric-like phase by the Y-doping cycles owing to the high proportion of the t-phase by ZrO_2_ in the initial HZO thin films [1]. It is interesting to note that H4Z6 film had ferroelectric properties until Y-doped 2 cy and showed antiferroelectric-like properties after 4 cy Y doping, whereas the other samples still retained ferroelectric properties. This means that the Y concentration and the Hf and Zr composition ratio are independent variables for the change of the dielectric constant. Thus, it was confirmed that the boundary between ferroelectric and antiferroelectric-like properties of Y-HZO thin films can be changed by the phase transition not only the Y-doping cycle but also the initial composition of HZO thin films. Figure 4c shows the variation in the 2E_c_ (coercive field) value with the Y-doping cycle for the as-deposited Y-HZO thin films with various compositions. The 2E_c_ value tends to decrease with an increasing number of Y-doping cycles in the case of Y-H5Z5 and Y-H4Z6. For Y-H6Z4, the 2E_c_ value decreased after Y-doping cycle of 2. This is because the energy required for polarization switching decreases with the dominant phase transition occurring in the order of the t-phase and c-phase upon Y doping. This reduction in the 2E_c_ value of the ferroelectric thin films is expected to facilitate a lower driving field requirement to maintain the 2P_r_ value, ultimately leading to a decrease in the probability of breakdown [13,29,30,31].

Figure 5a,b shows the dielectric constant as a function of Y-doping cycle for as-deposited Y-H6Z4 and Y-H4Z6 thin films in the range of ±3 V, respectively. The curve shapes of both Y-HZO thin films continuously changed with the Y-doping cycles, which was reported to be an inherent feature originating from the dielectric properties of the films. Thus, change in dielectric behavior of Y-HZO thin films with phase transition due to Y-doping shown in *P-E* curves of Figure 4a was further demonstrated by the ε_r_-V curves. Figure 5c shows the variation in the dielectric constant at +3 V with the Y-doping cycle for the Y-H6Z4 and Y-H4Z6 thin films. As the number of Y-doping cycles increased to seven, the dielectric constant increased from 15 to 28 for the Y-H6Z4 thin films and from 21 to 36 for the Y-H4Z6 thin films. The correlation between the dielectric constant and Y-doping cycle confirmed that the crystalline phase of the Y-HZO thin film is affected by the Y dopant [7,32]. More specifically, Y doping effectively enhanced the dielectric constant of the Y-HZO thin film by suppressing the m phase and inducing the formation of o-and t-phases. In contrast, after 10 cycles of Y doping, the dielectric constant decreased sharply for both compositions. This was considered to be the result of complete crystallization of the o- or t-phase in less than 10 cycles after 7 cycles of Y doping, followed by stabilization of the c-phase by a phase transition at doping concentrations above 10 cycles [7]. The decreased dielectric constant is similar to the previously reported dielectric constants of 25 and 32 for the c-phases of HfO_2_ and ZrO_2_, respectively [33]. The phenomenon of c-phase stabilization due to Y doping can be supported by the paraelectric properties in the *P*-*V* curve of Figure 4a and the flat *C-V* shape curve of Figure 5a,b. The variation in the leakage current density measured at +3 V with respect to the Y-doping cycles for the as-deposited Y-HZO thin films is shown in Figure 5d. The leakage current characteristics of both the Y-H6Z4 and Y-H4Z6 thin films were also improved by Y doping compared with those of the pristine HZO thin films. The pristine Y-H6Z4 and Y-H4Z6 thin films exhibited leakage current densities of 8.66 × 10^−6^ A/cm^2^ and 1.28 × 10^−5^ A/cm^2^. After 4 cycles of Y-doping for Y-H6Z4 and 7 cycles of Y-doping for Y-H4Z6, leakage current characteristics improved by 13.7 times (8.66 × 10^−6^ A/cm^2^ to 6.34 × 10^−7^ A/cm^2^_)_ and 27.3 times (1.28 × 10^−5^ A/cm^2^ to 4.68 × 10^−7^ A/cm^2^) compared to the pure state, respectively. This is because the doped Y combined with the oxygen vacancy to form a (V_O_-Y_Hf/Zr_)^+^ complex defect. By forming (V_O_-Y_Hf/Zr_)^+^ complex defects, the leakage current characteristics were improved by reducing the depth of the traps caused by oxygen vacancies located in the middle of the bandgap [32,34]. The enhanced leakage current characteristics are expected to affect the reliability of Y-HZO thin films.

### 3.5. Reliability Test of As-Deposited Y-HZO Thin Films

Finally, the effect of the improved electrical properties of the as-deposited Y-HZO thin films on device reliability was investigated. A fatigue endurance cycling test was performed on the Y(2cy)-H4Z6 thin film with the highest ferroelectric properties and excellent leakage current-density characteristics. For comparison, endurance tests of the H4Z6 thin films with and without 2 cycles of Y doping were conducted, the voltage of ±4 V was applied 10^10^ times to the TiN/Y-HZO/TiN capacitors, as shown in the schematic diagram of Figure 6a. Figure 6b,c shows the *P-V* curves of the initial H4Z6 and Y(2cy)-H4Z6 thin films before and after cycling, respectively. The hysteresis loops of both films remained stable even after the endurance cycling test, which was attributed to the absence of post thermal treatment process that deteriorated the leakage current density. This result can be seen in more detail in Figure 6d, which shows the variation in the 2P_r_ and 2E_c_ values with cycling for both the undoped and Y(2cy)-H4Z6 thin films. The undoped Y(2cy)-H4Z6 thin films displayed great endurance up to 10^7^ cycles, with stably maintained 2P_r_ and 2E_c_ values without a wake-up effect; however, hard breakdown occurred after that. The Y(2cy)-H4Z6 thin film showed excellent endurance up to 10^10^ cycles, with no wake-up effect and stable 2P_r_ and 2E_c_ values. Notably, the P_r_ value was consistently maintained without hard breakdown, even though the cycles were repeated 1000 times more than for the pure H4Z6 thin films. We conclude that this was due to the very low leakage current density caused by Y doping, in addition to the effect of the non-thermal treatment. Previous studies emphasized the tradeoff between the wake-up effect and endurance [10,13,35]. However, this study is significant in that it demonstrated both excellent endurance and wake-up-free characteristics even at high applied voltages without a trade-off between the wake-up effect and endurance, expanding the applicability of the as-deposited Y-HZO thin films as reliable devices.

## 4. Conclusions

We investigated the effects of Y-doping on the ferroelectric properties and additional electrical properties of as-deposited 10-nm-thick HZO thin films fabricated via ALD. The deposited Y-HZO films exhibited excellent crystallization without the need for post-thermal treatment. The films showed a phase transformation (ferroelectric–antiferroelectric–paraelectric) that varied based on the initial composition of HZO and the Y-doping cycle. Optimized Y-HZO thin films demonstrated improved ferroelectric properties compared to pure HZO thin films. Specifically, when 2 cy of Y was doped to H4Z6 thin film, which had the highest ferroelectricity in the pristine state, it had the largest ferroelectricity of 27.5 μC/cm^2^, and increased by about 1.5 times compared to the pristine state. Additionally, Y acted as a passivating agent for oxygen vacancies, resulting in reduced leakage current density in the Y-HZO thin films and excellent endurance. The optimized Y-HZO thin films exhibited stable ferroelectric performance without a significant increase in P_r_ or fatigue for up to 10^10^ cycles. In contrast, the pure HZO thin films experienced hard breakdown at only 10^7^ cycles. These findings establish the conditions for fabricating thin films with excellent electrical properties based on the various dielectric properties, taking into account the initial composition of HZO thin films and the Y-doping cycle. Therefore, this research contributes to the improvement of device characteristics of HZO ferroelectric devices such as FRAM, FeFET, and FTJ by not only enhancing the ferroelectric properties but also overcoming the endurance limitations of HZO thin films.

## Figures and Tables

**Figure 1 nanomaterials-13-02187-f001:**
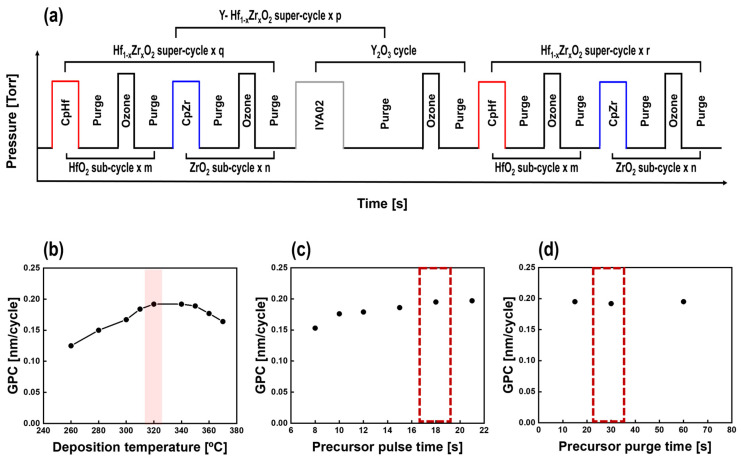
(**a**) Schematic of the ALD cycle for deposition of Y-HZO thin films. Growth per cycle (GPC) of Y_2_O_3_ thin films as a function of (**b**) deposition temperature, (**c**) IYA02 precursor pulse time, and (**d**) IYA02 precursor purge time.

**Figure 2 nanomaterials-13-02187-f002:**
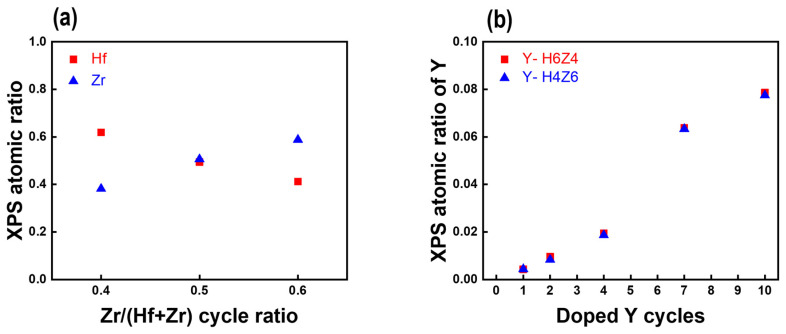
X-ray photoelectron spectroscopy (XPS) atomic ratio of (**a**) Hf and Zr according to Zr/(Hf + Zr) cycle ratio and (**b**) Y according to Y-doping cycle.

**Figure 3 nanomaterials-13-02187-f003:**
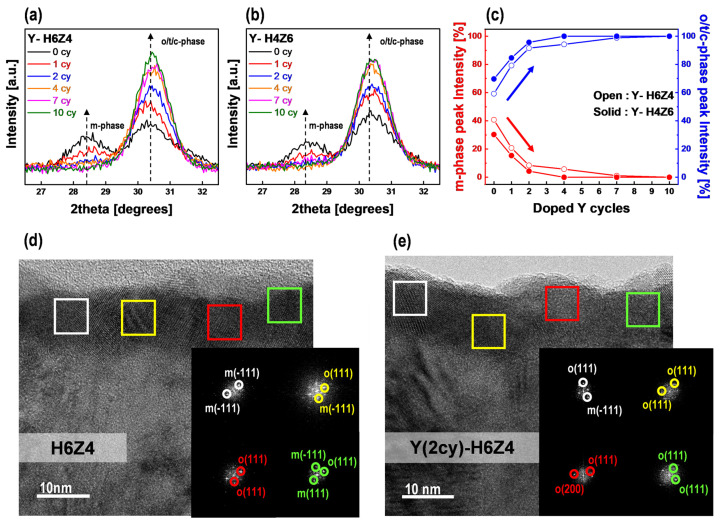
Grazing-angle incidence X-ray diffraction (GI-XRD) spectra of (**a**) as-deposited Y-H6Z4 thin films and (**b**) as-deposited Y-H4Z6 thin films according to Y-doping cycle. (**c**) Variation of the intensity ratio of the monoclinic phase (m-phase) peak and orthorhombic phase (o-phase)/tetragonal phase (t-phase)/cubic phase (c-phase) peak of as-deposited Y-H6Z4 and Y-H4Z6 thin films. TEM images of (**d**) as-deposited H6Z4 thin films and (**e**) as-deposited Y(2cy)-H6Z4 thin films. Inset: FFT diffraction pattern.

**Figure 4 nanomaterials-13-02187-f004:**
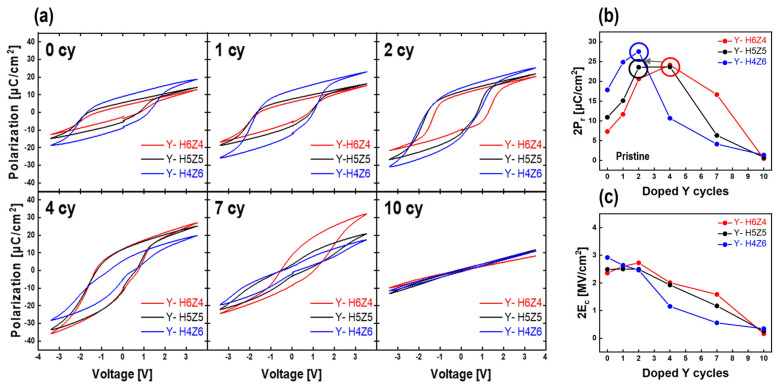
(**a**) Polarization-voltage (*P-V*) hysteresis curves of as-deposited Y-HZO thin films according to Y-doping cycle. (**b**) Variation of remanent polarization (2P_r_) values of as-deposited Y-HZO thin films according to Y-doping cycle. (**c**) Variation of coercive field (2E_c_) values of as-deposited Y-HZO thin films according to Y-doping cycle.

**Figure 5 nanomaterials-13-02187-f005:**
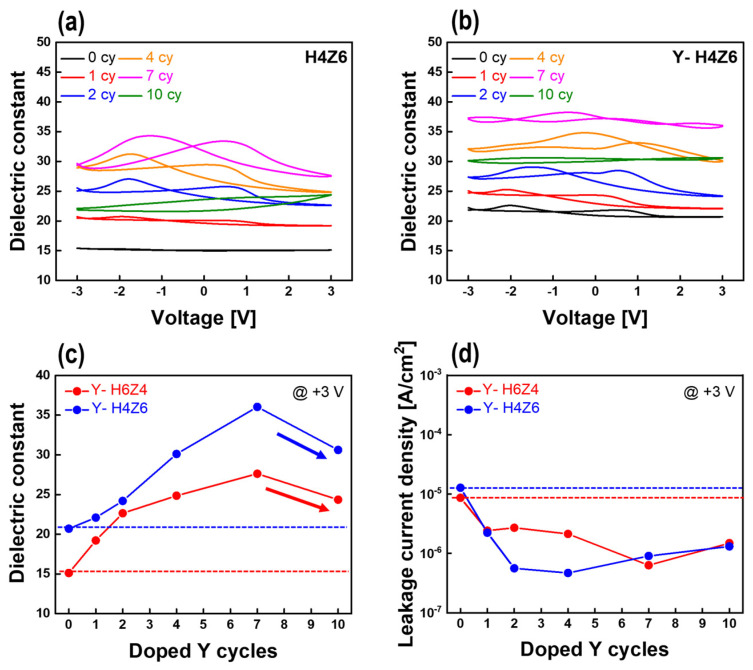
Dielectric constant—voltage (*ε_r_–V*) hysteresis curves of (**a**) H4Z6 thin films and (**b**) Y-H4Z6 thin films according to Y-doping cycle. Variation of (**c**) dielectric constant and (**d**) leakage current density of Y-HZO thin films a function of Y-doping cycle. (Dotted lines in (**c**,**d**) indicate the reference values of un-doped HZO films).

**Figure 6 nanomaterials-13-02187-f006:**
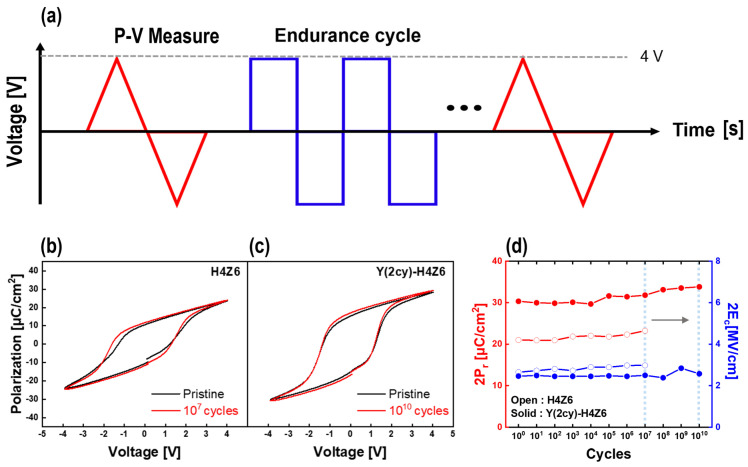
(**a**) Schematic of the pulse for the endurance test. Polarization-voltage (*P*-*V*) hysteresis curves of (**b**) as-deposited H4Z6 thin films for the pristine state and the state after 10^7^ cycles and (**c**) as-deposited Y(2cy)-H4Z6 thin films for the pristine state and the state after 10^10^ cycles. (**d**) Variation of remanent polarization (2P_r_) and coercive field (2E_c_) values according to endurance cycles.

## Data Availability

Not applicable.

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
