# Peer review of "Yttrium Doping Effects on Ferroelectricity and Electric Properties of As-Deposited Hf1−xZrxO2 Thin Films via Atomic Layer Deposition"

_nanomaterials, 2023, doi:10.3390/nano13152187_

Round 1

Reviewer 1 Report

In this study, experimenters investigated the effects of Y doping cycles on the ferroelectric and electrical properties of as-deposited Yttrium doping Hf1-xZrxO2 thin films with vaying compositions fabricated through ALD. The Y-HZO thin films were stably crystallized without the need for post-thermal treatment and exhibited transition behavior depending on the Y-doping cycle and initial composition ratio of the HZO thin films. These Y-HZO films have the advantages of enhanced dielectric constant, leakage current density and improved durability. The results shown that the optimized Y-doping cycle induces the phase transition of Y-HZO films, improves the ferroelectric properties of Y-HZO films, and exhibits stable fatigue-free behavior up to 1010 cycles. However, I considered it can be published in nanomaterials after a minor revision:

(1) The selection of the precursor purge time in Figure 1(d) by the researcher in the manuscript should be clearly explained, and the specific reason for the selection of 30s should be stated.

(2) In the manuscript, the researcher should give an analysis of the specific test data of XPS, and show how the atomic ratios of Hf and Zr were obtained from the specific data.

(3) In line 210 on page 6 of the manuscript, the researchers describe that the improvement of ferroelectricity is due to the formation of oxygen vacancies. I think specific explanations need to be given here.

(4) In line 230 on page 7 of the manuscript, the authors describe a tendency for the 2Ec value to decrease as the number of yttrium doping cycles increases. This is somewhat different from the actual trend in the figure. I think both the language and grammar in the article need to be improved. add some references such as “DOI: 10.1002/smll.202303463., Materials Today Physics, 2022, 29: 100902., Journal of Colloid and Interface Science, 2023, 634: 268-278.”

(5) In the manuscript, researchers should refer to a large number of documents to more specifically explain the research content. In addition, the format and size of the charts in the paper need to be adjusted to look better.

minor 

Reviewer 2 Report

1. The Figure1(a) diagram is too far apart from the text description and do the Time and Pressure in the picture have units? Are the lengths and heights in the pictures drawn from specific test results? What are the measurement and evaluation criteria? All need to be clearly described in the text;

2. The Figure3(a) and (b) are connected together in such a way that the data results for the two materials are not clearly distinguishable and need to be split into two separate data plots;

3. The voltage and time in Figure6(a) are missing units;

4. Both in the introduction and in the conclusion, Y-HZO is compared with pure HZO, but there is no relevant data in the test and discussion to support these conclusions, and relevant comparison data needs to be addedï¼›

5. What are the specific criteria for evaluating the performance of HZO films? Ferroelectric properties? Additional electrical properties? What directions we can take to improve the performance through the results that are worthwhile, application prospects, etc. can be added.

6. Figure 5 could do with a bit of typographical adjustment, 2 x 2 combination of images, with appropriate enlargement of each image.

7. Please double check the punctuation in the text, e.g. there is no full stop in this section of 3.5.

8. In section 3.2 for the chemical composition of Y-HZO to study, doping with different content of Hf and Zr, has a great impact on the film properties, so what is the optimal doping amount, can be added in this section.

9. The abbreviations of the figure labels need to be harmonised throughout the text. In section 3.1, "fig 1b" should be changed to "Figure 1b", please make the change throughout the text.And, the presence or absence of an apostrophe between Figure and number should also be standardized throughout. For example, Figure 1c is different from Figure. 1d in format.

1. The Figure1(a) diagram is too far apart from the text description and do the Time and Pressure in the picture have units? Are the lengths and heights in the pictures drawn from specific test results? What are the measurement and evaluation criteria? All need to be clearly described in the text;

2. The Figure3(a) and (b) are connected together in such a way that the data results for the two materials are not clearly distinguishable and need to be split into two separate data plots;

3. The voltage and time in Figure6(a) are missing units;

4. Both in the introduction and in the conclusion, Y-HZO is compared with pure HZO, but there is no relevant data in the test and discussion to support these conclusions, and relevant comparison data needs to be addedï¼›

5. What are the specific criteria for evaluating the performance of HZO films? Ferroelectric properties? Additional electrical properties? What directions we can take to improve the performance through the results that are worthwhile, application prospects, etc. can be added.

6. Figure 5 could do with a bit of typographical adjustment, 2 x 2 combination of images, with appropriate enlargement of each image.

7. Please double check the punctuation in the text, e.g. there is no full stop in this section of 3.5.

8. In section 3.2 for the chemical composition of Y-HZO to study, doping with different content of Hf and Zr, has a great impact on the film properties, so what is the optimal doping amount, can be added in this section.

9. The abbreviations of the figure labels need to be harmonised throughout the text. In section 3.1, "fig 1b" should be changed to "Figure 1b", please make the change throughout the text.And, the presence or absence of an apostrophe between Figure and number should also be standardized throughout. For example, Figure 1c is different from Figure. 1d in format.

Reviewer 3 Report

Ms. Ref. No.: nanomaterials-2494935

Title: "Yttrium Doping Effects on Ferroelectricity and Electric Properties of As-deposited Hf1-xZrxO2 Thin Films via Atomic Layer Deposition"

In this study, the authors investigated the effects of Y-doping on the ferroelectric properties and additional electrical properties of as-deposited 10-nm-thick HZO thin films fabricated through atomic layer deposition. The paper is really interesting, well conducted, and fits the objectives of the journal; but it is necessary to review some points in order to improve the quality of the paper:

- The first keyword is usually capitalized.

- The writing needs to be polished. The reviewer suggests rewriting the long sentences in the introduction and conclusion parts into short ones to make them easy to be understood.

- The authors did not explain the novelty and significance of their work in the introduction part. Make sure that the novelty of this manuscript is bolded.

- There are some formatting mistakes in the references section, I suggest the authors check and correct them. There are incomplete references or erroneous data, others with typos in the journal name, or chemical formulae in the title, for example, ref #20.

- How did authors measure and control the thickness of ALD? Just by TEM micrographs? Was the layer uniform?

- The authors stated that “Thus, it was confirmed that the boundary between ferroelectric and antiferroelectric-like properties of Y-HZO thin films can be changed by the phase transition not only the Y-doping cycle but also the initial composition of HZO thin films.” Please give more explanation about this matter in the text.

- I suggest the authors incorporate some sentences of future perspectives related to the topic in the conclusion section. 

Minor editing of English language required!

Round 2

Reviewer 3 Report

The authors corrected the paper following the reviewer's advice and improved the quality of the manuscript. It can be accepted as it is. 

Minor editing of English language required!